# Icaritin Exerts Anti-Cancer Effects through Modulating Pyroptosis and Immune Activities in Hepatocellular Carcinoma

**DOI:** 10.3390/biomedicines12081917

**Published:** 2024-08-21

**Authors:** Yuanyuan Jiao, Wenqian Li, Wen Yang, Mingyu Wang, Yaling Xing, Shengqi Wang

**Affiliations:** 1College of Traditional Chinese Medicine, Tianjin University of Traditional Chinese Medicine, Poyanghu Road, Jinghai District, Tianjin 301617, China; jiaoyy1995@163.com; 2Bioinformatics Center of AMMS, Taiping Road, Haidian District, Beijing 100850, China; joice09876q@163.com (W.L.); 18363081136@163.com (W.Y.); wangmeo12138@163.com (M.W.); 3College of Traditional Chinese Medicine, Shandong University of Traditional Chinese Medicine, Daxue Road, Jinan 250355, China

**Keywords:** icaritin, pyroptosis, GSDMD, GSDME, immunotherapy, hepatocellular carcinoma

## Abstract

Icaritin (ICT), a natural compound extracted from the dried leaves of the genus Epimedium, possesses antitumor and immunomodulatory properties. However, the mechanisms through which ICT modulates pyroptosis and immune response in hepatocellular carcinoma (HCC) remain unclear. This study demonstrated that ICT exhibits pyroptosis-inducing and anti-hepatocarcinoma effects. Specifically, the caspase1-GSDMD and caspase3-GSDME pathways were found to be involved in ICT-triggered pyroptosis. Furthermore, ICT promoted pyroptosis in co-cultivation of HepG2 cells and macrophages, regulating the release of inflammatory cytokines and the transformation of macrophages into a proinflammatory phenotype. In the Hepa1-6+Luc liver cancer model, ICT treatment significantly increased the expression of cleaved-caspase1, cleaved-caspase3, and granzyme B, modulated cytokine secretion, and stimulated CD8^+^ T cell infiltration, resulting in a reduction in tumor growth. In conclusion, the findings in this research suggested that ICT may modulate cell pyroptosis in HCC and subsequently regulate the immune microenvironment of the tumor. These observations may expand the understanding of the pharmacological mechanism of ICT, as well as the therapy of liver cancer.

## 1. Introduction

In 46 countries worldwide, liver cancer is not only highly prevalent but also ranks among the top three causes of cancer-associated deaths. Based on the Global Cancer Observatory, liver cancer was diagnosed in 905,700 individuals in 2020, resulted in 830,200 deaths. It is hypothesized that there will be a 55% increase in both new cases and mortalities attributed to liver cancer by 2040 [1]. In China, liver cancer accounts for 45.3% of the global incidence and 47.1% of the global mortality [2]. The primary causes of liver cancer are hepatitis B virus infection and cirrhosis in China [3]. By the time a diagnosis is made, most patients are in advanced stages. Thus, the search for safer and more effective drugs is of great significance for liver cancer therapy, as it can improve patient outcomes and quality of life.

Multiple studies have reported that promoting pyroptosis in tumor cells is a novel antitumor therapy strategy [4,5,6,7,8]. Pyroptosis regulates the tumor microenvironment (TME), reducing tumor immune escape and tumorigenesis [9,10,11,12]. Pyroptotic cells emit a significant quantity of inflammatory cytokines, attracting immune cells and ultimately aiding in the elimination of the tumor [13,14]. In HCC cells, NK cells and CD8^+^ Cytotoxic T Lymphocytes (CTLs) can secret granzymes (Gzms) that activate gasdermins (GSDMs) to induce pyroptosis [15,16,17]. GzmA induces pyroptosis through GSDMB, while GzmB directly induces pyroptosis via Gasdermin E (GSDME) and can also indirectly amplify pyroptosis through caspase-3 [18,19].

Pyroptosis, discovered around 1992, is an inflammatory cell death pathway that is gaining significant attention for its critical role in the innate immune response [20,21]. Gasdermins (GSDMs) were first reported in 2007, and include GSDM-A, -B, -C, -D, and -E, and Pejvakin (also known as DFNB59) [22]. These proteins share structural similarities but differ in function and expression. GSDM-A, -B, -C, -D, and -E are primarily involved in pyroptosis. They each possess an N-terminal (NT) domain that forms pores and a C-terminal inhibitory domain. The cleavage by caspases at interdomains enables the NT domain to bind to membrane phospholipids, forming pores around 18 nm in diameter, resulting in the release of interleukin (IL)-1β and IL-18 [23,24]. For example, GSDMD is cleaved and activated by caspase-1/4/5/11 [25], while GSDME is cleaved and activated by caspase-3 [23,26,27]. In contrast, hPJVK is predominantly expressed in the inner ear and certain neurons, playing a crucial role in auditory hair cell function and survival, and it is not directly associated with pyroptosis [28,29].

Numerous active components derived from traditional Chinese medicine (TCM), known for their efficacy, minimal adverse reactions, and the ability to modulate immune function, have been extensively applied in antitumor therapy [30,31]. Icaritin (ICT), a highly potent antitumor compound, is derived from the dried leaves of several Epimedium species, all of which are TCMs [32,33]. ICT is renowned for its anti-inflammatory and immune-modulatory properties across different pathological conditions [34]. Pharmacological experiments have demonstrated the efficacy of ICT in suppressing tumor cell growth in lung [35], breast [36], liver [37], and prostate cancers [38]. A clinical trial revealed that ICT was safe and extended survival in advanced liver cancer patients, primarily due to its immunomodulatory functions [32]. Earlier research has shown that ICT reduces the viability of HCC cells by triggering apoptosis [39,40]. However, the contribution of pyroptosis to the antitumor effects of ICT remains unclear. This study focuses on the function of ICT in cell pyroptosis in both hepatocellular carcinoma (HCC) cells and mouse models, uncovers the regulation of TME, and ultimately exposes the mechanisms used by ICT in the treatment of liver cancer.

## 2. Materials and Methods

### 2.1. Compounds and Antibodies

ICT with a purity of 98% (CAS: 118525-40-9) was obtained from Beijing Shenogen Pharma Group (Beijing, China). Z-VAD-FMK (C1202), Ac-DEVD-CHO (C1206), Cleaved Caspase-3 (AC033) Rabbit pAb, and an Annexin V-FITC apoptosis detection kit (C1062) were procured from Beyotime (Beijing, China). Disulfiram (HY-B0240) was sourced from MCE (Shanghai, China). Antibodies against Gasdermin D (E9S1X), F4/80 (D2S9R), PD-L1 (D5V3B), and CD8α (D4W2Z) were purchased from CST (Danvers, MA, USA). The recombinant monoclonal antibody for DFNA5/GSDME-N-terminal (ab215191) was acquired from Abcam (Cambridge, UK). Cleaved-Caspase 1 Rabbit pAb (341030) was sourced from Zenbio (Chengdu, China). Recombinant Anti-beta actin (GB15003-100) and HRP-conjugated Rabbit Anti-Goat IgG (GB23204) antibodies were procured from Servicebio (Wuhan, China). The CCK-8 cell counting kit (A311) was obtained from Vazyme (Nanjing, China). The CytoTox 96^®^ Non-Radioactive Cytotoxicity Assay (G1780) was acquired from Promega (Beijing, China). RNA Easy Fast Tissue/Cell (DP451), Fasting gDNA SuperMix (KR118), and Talent qPCR PreMix kits (FP209) were obtained from TIANGEN (Beijing, China). The Bio-Plex Pro Mouse Cytokine 23-plex Assay (AB_2857368) was purchased from Bio-Rad Laboratories (Hercules, CA, USA). Lipofectamine 3000 reagent (L3000015) was sourced from Thermo (Waltham, MA, USA) and JetPRIME^®^ reagent (101000046) from Polyplus (Illkirch, France). The TSA Plus fluorescence kit (M60009RDPD) was obtained from Akoya Biosciences (Marlborough, MA, USA).

### 2.2. Colony Formation Assay

In six-well plates, HepG2 and Huh7 cells were seeded at 5000 cells/well. After the cells adhered, they were exposed to ICT at concentrations of 0, 20, and 40 µM for 48 h. Following this, the cells were incubated without ICT for an additional 10–14 days. The colonies were washed twice with PBS, fixed with 4% paraformaldehyde (PFA), and stained with 0.1% crystal violet. Finally, images of the colonies of HepG2 and Huh7 cells were recorded.

### 2.3. Cytotoxicity Assays

To determine the viability of ICT-treated HepG2, MHCC97H, HCCLM3, Huh7, and LO2 cells, we treated HepG2, MHCC97H, HCCLM3, and Huh7 cells with diverse concentrations of ICT (0, 5, 10, 15, 20, 25, 30, 40, 50, 60 µM) for 24 and 48 h, respectively. LO2 cells were exposed to ICT (0, 5, 10, 20, 40, 60, 80, and 100 µM). After incubating the cells with CCK8 for an additional 30 min, the absorbance at 450 nm was surveyed using a BioTek Epoch microplate reader (Winooski, VT, USA).

### 2.4. LDH Release Assay

To determine the release of Lactate dehydrogenase (LDH) induced by ICT, HepG2, MHCC97H, Huh7, and HCCLM3 cells were exposed to different concentrations of ICT (0, 20, 30, 40, 50 µM) for 24 and 48 h. The Maximum LDH Release Control (total LDH release) was set up by adding 10% Triton X-100 (2 μL per 100 μL) to the control group of cells [23]. The mixture was then incubated for 10–15 min, and then the samples were collected for LDH detection. The medium was collected after a predetermined period. The wells were shielded from light and incubated at 37 °C for 10–30 min, followed by surveying the absorbance at 490 nm [41]. LDH release(%) = (ICT-treatment LDH release − Control LDH release)/(Total LDH release − Control LDH release) × 100%.

### 2.5. Annexin V-FITC/PI Staining

Fluorescence microscopy and flow cytometry were employed to assess pyroptosis in HepG2 cells. Cells were exposed to ICT (0, 12.5 and 25 µM) for a duration of 48 h, then Annexin V-FITC and propidium iodide (PI) were used for staining. After 15 min of incubation in the dark, fluorescence microscopy was utilized to observe both the quantity and morphology of pyroptotic cells. HepG2 cells were differentiated into live, necrotic, and apoptotic cells by flow cytometry, analyzed with FlowJo v10.8.1 software.

### 2.6. Pyroptosis-Related Inhibitors

HepG2 and Huh7 cells were cultured to approximately 80% confluence before simultaneous exposure to 25 µM ICT and a series of inhibitors for 48 h. The inhibitors used included 20 µM Z-VAD-FMK (a pan-caspase inhibitor), 5 µM disulfiram (a GSDMD inhibitor), and 20 µM Ac-DEVD-CHO (a caspase-3 inhibitor). Following ICT treatment, LDH release was measured.

### 2.7. Western Blotting

HepG2 cells were exposed to diverse concentrations of ICT (0, 12.5, and 25 µM) for 48 h, then lysed in RIPA buffer at 4 °C for 30 min, followed by measurement of protein concentrations. A quantity of 40 µg total protein was separated on 12% SDS-PAGE gels and transferred onto PVDF membranes. Subsequently, the membranes were blocked with 5% non-fat milk for 45 min. They were incubated overnight at 4 °C with antibodies (dilution ratio of 1:1000) against cleaved-caspase1, GSDMD-NT, cleaved-caspase3, and GSDME-NT. Then, they were treated with HRP-conjugated Rabbit Anti-Goat IgG for 2 h. Finally, E-blot Touch Imager (E-blot, Shanghai, China) was used to visualize immunoreactive proteins.

### 2.8. Co-Culture of THP-1 Cells with HepG2 Cells

The transwell insert was placed in a 12-well plate, and each well was divided into upper and lower compartments. HepG2 cells (1.3 × 10^5^ cells/mL) were seeded in the upper compartments, whereas THP-1 cells (3 × 10^5^ cells/mL) were seeded in the lower compartments. THP-1 cells were differentiated by exposure to 20 ng/mL of phorbol-12-myristate-13-acetate (PMA) for 12 h. Following differentiation, co-cultivation with HepG2 cells was commenced immediately. After co-culturing for 48 h, the cells were concurrently treated with diverse concentrations of ICT (0, 2, 4, and 6 µM), following which the macrophages and medium were harvested for further analysis.

### 2.9. Quantitative PCR Analyses

RNA was isolated and evaluated before being synthesized as first-strand cDNA. The reaction mixture stored at 42 °C for 15 min, then denatured at 95 °C for 3 min, generating cDNA. RT-PCR was conducted with an initial denaturing step at 95 °C for 3 min, followed by 40 cycles of the amplification (5 s at 95 °C and 15 s at 60 °C) [41]. mRNA levels of COX-2, IL-6, CD163, IL-10, and TNF-α were quantified using the 7500 Fast RT-PCR Instrument. The ∆∆CT-values were calculated, normalized to β-actin expression, and compared to the control sample. Table 1 provides primer sequences.

### 2.10. ELISA

The THP-1 cell culture medium’s IL-1β, L-6, and IL-18 levels were measured. In basic terms, the medium and biotinylated antibodies were applied to microplates. Following one to three hours of waiting, wash buffer was used to clean the wells. Soon after incubating with Streptavidin-HRP for 30 min, the wells were washed. 3,30,5,50-Tetramethylbenzidine (TMB) substrate was added, and the absorbance at 450 nm was determined. Using a standard curve, the levels of cytokines were calculated.

### 2.11. Cell Transfection and siRNA Knockdown

GSDMD and GSDME overexpression plasmids were purchased from General Biotechnology (Chuzhou, China). Cultured in 12-well plates, HepG2 cells attained roughly 80% confluence and underwent transient transfection. For GSDMD and GSDME overexpression, 1 µg of plasmid was transfected into cells with Lipofectamine™ 3000 reagent based on the manufacturer’s directions. For siRNA GSDMD and GSDME knockdown, 50 nmol of siRNAs was transfected into HepG2 cells with JetPRIME^®^ reagent. After overexpression and siRNA knockdown of GSDMD/E, transfected cells were exposed to 20 µM ICT for 48 h. The efficacies of overexpression and knockdown were assessed using Western blotting. The sequences of siRNA GSDMD and GSDME are listed in Table 2.

### 2.12. Orthotopic Liver Cancer Mice Model

Six-week-old male WT C57BL/6N mice were anesthetized using a 1% pentobarbital solution. The mice were positioned supine on the operating table and the left lobe of the liver was gently exposed. Subsequently, 1 × 10^7^ cells/mL of Hepa1-6+Luc cells suspended in Corning Matrigel (Matrigel:DMEM with 10% fetal bovine serum = 1:1) was injected in a volume of 30 μL into the left lobe of the liver [42]. Sterile cotton swabs were used to staunch any bleeding, and the mouse’s abdomen was closed with sutures. ICT was administered via oral gavage at a dosage of 70 mg/kg twice daily for 17 days (n = 8) (approval no. IACUC-DWZX-2020-667).

### 2.13. H&E Staining

Hematoxylin and Eosin (H&E) staining was conducted to histologically examine the liver tissue samples. After 24 days of modeling, tissue sections were fixed in 4% PFA, dehydrated, and embedded in paraffin blocks. Histopathological evaluation involved staining sections of 5 μm thickness with H&E. Each experimental group was represented by three liver sections for analysis. After staining, examination and image capture followed (n = 3).

### 2.14. Immunohistochemistry

The protein level of cleaved-caspase1, cleaved-caspase3, and granzyme B were detected via immunohistochemistry. At 24 days post-modeling, the left lobes of harvested liver tissues were fixed in 4% PFA. They were then incubated overnight with cleaved-caspase1 antibody (1:100 dilution), cleaved-caspase3 antibody (1:50 dilution), and granzyme B antibody (1:500 dilution) at 4 °C. The next day, the tissues were treated with goat anti-rabbit secondary antibody (1:100 dilution) at 37 °C for 40 min [43]. Three liver sections were examined from each group. Positive immunohistochemical staining was brown in the 10× visual field (n = 3).

### 2.15. Multiplex Immunohistochemistry

Beyond a 24-day modeling period, the liver tissues’ left lobes retained their integrity in 4% PFA. Upon completion, tissues were washed with xylene and graded alcohol to deparaffinize and rehydrate them. In a 10 mM sodium citrate solution (pH 6), antigen was recovered by boiling for 15 min. Sections were blocked for one hour and then treated with PD-L1 (1:400), F4/80 (1:400), and CD8α (1:500) antibodies for the entire night at 4 °C. Following an hour of incubation with secondary antibodies coupled with HRP, TSA-dendron fluorophores were used for detection. Multiplex immunohistochemical fluorescence observations were performed using a digital whole-slide scanning system at 2× magnification (n = 3).

### 2.16. Bio-Plex Multiplex Mouse Cytokine 23-Plex Assay

The Bio-Plex Cytokine Assay Kit was used to examine 23 different cytokines in mouse plasma. Twenty-three cytokines were examined in plasma samples from the ICT group (n = 8) and the HCC group (n = 8) in accordance with the manufacturer’s instructions. The cytokines included Eotaxin, G-CSF, GM-CSF, IFN-γ, IL-1α, IL-1β, IL-2, IL-3, IL-4, IL-5, IL-6, IL-9, IL-10, IL-12 (p40), IL-12 (p70), IL-13, IL-17A, KC, MCP-1 (MCAF), MIP-1α, MIP-1β, RANTES, and TNF-α. Student’s *t*-test was used to evaluate the concentrations of cytokines.

### 2.17. Statistical Analysis

The mean ± standard deviations (SD) are used to display data. GraphPad Prism 8.0 was used for the statistical analysis, and a one-way ANOVA was used for multiple comparisons. Levels of significance are indicated as follows: * *p* < 0.05, ** *p* < 0.01, *** *p* < 0.001.

## 3. Results

### 3.1. ICT Inhibited Survival and Proliferation of HCC Cells

The chemical structure of ICT, which was extracted from dried leaves of the genus Epimedium, is displayed in Figure 1A. Evaluating the vitality and proliferation of HCC cells allowed us to assess the anticancer effects of ICT. HepG2 and Huh7 cells given ICT showed a decrease in the quantity of cells during long-term colony formation trials (Figure 1B). Furthermore, ICT reduced the survival of HepG2, Huh7, MHCC97H, and HCCLM3 cells in a dosage- and time-dependent manner, according to the CCK8 cell counting test. On the other hand, LO2 cells from human hepatocytes treated with ICT at doses between 0 and 40 µM were more viable than HCC cells (Figure 1C). To sum up, the findings indicate that ICT considerably reduces the ability of HCC cells to survive and proliferate.

### 3.2. ICT Induced Pyroptosis in HCC Cells

In order to illuminated the influence of ICT on the process of pyroptosis, HepG2 and Huh7 cells were subjected to a treatment of 20 μM ICT, while MHCC97H and HCCLM3 cells underwent a treatment of 35 μM ICT, both for 24 and 48 h. The ICT treatment group resulted in a balloon-like appearance of the cells compared to the control group, demonstrating characteristic pyroptotic cell morphology (the red arrows indicate pyroptotic cells; Figure 2A). When cells undergo pyroptosis, the cell membrane ruptures and leaks the contents. LDH release is used as a pyroptosis marker due to its stable enzymatic activity [4,23]. Therefore, the release of LDH from HepG2, MHCC97H, HCCLM3, and Huh7 cells was assessed. It was observed that LDH release increased with increasing ICT concentrations (Figure 2B). With increasing ICT concentrations, an increase in the number of cells exhibiting Annexin V-FITC and PI double-positive staining was observed. Representative pyroptosis images showed cells swelling with large bubbles (Figure 2C). While 12.5 μM ICT-treated HepG2 cells were mainly Annexin V-FITC single-positive, 25 μM ICT-treated HepG2 cells proceeded directly into double-positive stage. This confirmed the novel role of ICT in a switch from apoptosis to pyroptosis (Figure 2D). Z-VAD-FMK (pan-caspase inhibitor), Ac-DEVD-CHO (caspase-3 inhibitor), and disulfiram (GSDMD inhibitor) were used to investigate whether ICT induces pyroptosis. Compared to the 25 μM ICT group, co-treatment with these inhibitors reduced LDH release in HepG2 cells (Figure 2E). These findings suggest that ICT triggers pyroptosis in HCC cells.

### 3.3. The Caspase1-GSDMD and Caspase3-GSDME Pathways Are Involved in ICT-Triggered Pyroptosis

The critical mechanisms underlying pyroptosis involve the activation of caspases and subsequent cleavage of GSDMs. In this study, the expression levels of cleaved-caspase1, GSDMD-NT, cleaved-caspase3, and GSDME-NT were assessed in HepG2 cells using Western blotting. ICT treatment groups resulted in a dose-dependent increase in cleaved-caspase1, cleaved-caspase3, GSDMD-NT, and GSDME-NT (Figure 3A). To confirm the functions of GSDMD and GSDME in ICT-induced pyroptosis, we employed overexpression and siRNA knockdown strategies. SiRNA-mediated knockdown of GSDMD and GSDME reduced LDH release (Figure 3B). Conversely, overexpression of GSDMD and GSDME enhanced ICT-induced LDH release and augmented the pyroptotic morphological features, indicating increased pyroptosis (Figure 3C). Additionally, the pan-caspase inhibitor (Z-VAD-FMK) decreased the activation of GSDMD-NT and GSDME-NT (Figure 3D). These findings suggest that caspases, GSDMD, and GSDME are critical for ICT-induced pyroptosis in HepG2 cells. Thus, caspase1-GSDMD and caspase3-GSDME pathways play an important role in ICT-triggered pyroptosis.

### 3.4. ICT-Induced Polarization of Tumor-Associated Macrophages In Vitro

HepG2 and THP-1 cells were co-cultured in order to study the relationship between immune and tumor cells in the TME (Figure 4A). The outcomes demonstrated that LDH release was enhanced by ICT treatment in a dose-dependent manner (Figure 4B). This implies that in the co-culture system, ICT facilitated pyroptosis. The inflammatory cytokines IL-1β, IL-6, and IL-18 were found to be increased by ELISA (Figure 4C). Inflammatory factors released during caspase-1-GSDMD-induced pyroptosis include IL-18 and IL-1β. Furthermore, the results of qPCR demonstrated an increase in the mRNA expression of the proinflammatory (M1 type) macrophage markers (COX-2, IL-6, and TNF-α), but there was no discernible change in the expression of the anti-inflammatory (M2 type) markers (CD163 and IL-10) (Figure 4D). These results imply that ICT promoted pyroptosis in the co-culture system, promoting the release of inflammatory cytokines and a shift toward a proinflammatory macrophage phenotype, potentially enhancing antitumor effects.

### 3.5. ICT Inhibits Tumor Growth by Regulating Pyroptosis and Immune Response In Vivo

To explore the effectiveness of ICT against tumors in vivo, an orthotopic Hepa1-6+Luc cell liver cancer mouse model was used to reproduce the main pathological features of human HCC [44,45]. After treatment with ICT (70 mg/kg, twice daily for 17 days), the livers of mice were collected. ICT treatment significantly reduced tumor volume, in contrast with the control group (Figure 5A). H&E staining results showed significant hepatic interstitial infiltration of inflammatory cells (yellow arrowheads) in the ICT group compared to the HCC group (Figure 5B). Immunohistochemistry revealed increased expression of cleaved-caspase1, cleaved-caspase3, and granzyme B in the ICT treatment group (Figure 5C; brown precipitates indicate positive expression). This suggests ICT-induced pyroptosis in mouse paracarcinoma tissues. The cytokine detection results showed that IL-6, G-CSF, KC, and other proinflammatory factors significantly decreased in the ICT-treated group, suggesting that ICT can influence the antitumor immune response by affecting cytokine secretion (Figure 5D). Additionally, there was a rise in CD8+ CTL counts, whereas PD-L1 expression was downregulated in the ICT treatment group (Figure 5E). These findings indicate that ICT could enhance pyroptosis-related protein expression, regulate cytokine secretion, and promote immune cell infiltration.

## 4. Discussion

This study observed the role of ICT in inducing pyroptosis in HCC cells, both in vivo and in vitro, expanding the understanding of the biological processes of ICT in the therapy of liver cancer.

This study observed that 20–40 μM ICT can significantly inhibit the survival and proliferation of HCC cells, while having minimal impact on the viability of human hepatocyte LO2 cells. This indicates that ICT exhibits anti-hepatocarcinoma effects in vitro. Interestingly, typical pyroptotic changes, such as cell swelling with balloon-like structures originating from the plasma membrane, were observed in ICT-treated HepG2, Huh7, MHCC97H, and HCCLM3 cells. Pyroptosis affects cell membrane integrity, causing the escape of pro-inflammatory cytokines and LDH [46]. Increased ICT concentrations correlated with elevated LDH levels and a higher number of HepG2 cells exhibiting Annexin V-FITC and PI double-positive staining. Flow cytometry showed that HepG2 cells treated with 12.5 μM ICT exhibited Annexin V-FITC positivity, whereas 25 μM ICT treatment resulted in double positivity for Annexin V-FITC and PI, indicating a shift from apoptosis to pyroptosis with increasing ICT concentrations. These findings suggested that ICT induces pyroptosis in HCC cells.

Further study demonstrated the initiation of pyroptosis executors GSDMD and GSDME in HepG2 cells treated with ICT. The caspase1-GSDMD and caspase3-GSDME pathways are the mechanisms of ICT-induced pyroptosis. This is supported by overexpression and siRNA knockdown in HepG2 cells, which underscores the critical functions of GSDMD and GSDME in ICT-induced pyroptosis. Additionally, pan-caspase inhibitors, caspase-3 inhibitors, and GSDMD inhibitors significantly reduced LDH release in HepG2 cells. Pan-caspase inhibitors also decreased the expression of GSDME/D-NT, highlighting the importance of the caspase family in ICT-induced pyroptosis. These findings imply that ICT causes pyroptosis in HCC cells predominantly by promoting the GSDMD and GSDME pathways, which have been profoundly impacted by caspase activity.

The process of pyroptosis is crucial for regulating immune responses and inflammation. During pyroptosis, tumor cells secrete inflammatory cytokines that consist of IL-1β and IL-18, which attract macrophages and CTLs and promote tumor cell elimination [47,48,49,50]. This process polarizes macrophages towards a proinflammatory M1 phenotype, increasing the secretion of cytokines with anticancer properties such as TNF-α, IFN-γ, IL-1β, IL-6, and IL-18 [51,52,53]. To investigate the interaction between immune and tumor cells in the TME, HepG2 cells were co-cultured with THP-1 cells. The results showed that ICT induces pyroptosis, leading to the release of cytokines, likely IL-6, IL-1β, and IL-18. Furthermore, the mRNA expression of proinflammatory macrophage (M1) markers such as COX-2, IL-6, and TNF-α was increased, whereas that of anti-inflammatory (M2) markers did not show significant changes, indicating a shift in macrophages toward a proinflammatory phenotype. Therefore, ICT promoting pyroptosis in the TME and regulating macrophage polarization may be beneficial for tumor therapy.

In a hepa1-6+Luc liver cancer model, ICT treatment significantly reduced tumor size and promotes the infiltration of inflammatory cells in the hepatic interstitium. ICT upregulated the expression of cleaved-caspase1, cleaved-caspase3, and granzyme B in mouse para-carcinoma tissues. Additionally, ICT significantly reduced the expression of proinflammatory cytokines such as IL-6, G-CSF, and KC in mouse plasma. For example, IL-6 can promote tumor cell proliferation, invasion, and anti-apoptosis [54,55]. Reducing IL-6 expression may inhibit liver cancer progression and increase treatment sensitivity [32]. Multiplex immunohistochemistry revealed that ICT promoted the infiltration of CD8^+^ CTLs and downregulated PD-L1 expression in mouse para-carcinoma tissues. Thus, ICT may regulate cytokine expression levels, modulating inflammation, pyroptosis, and immune responses to inhibit tumor growth. Various immune cells can infiltrate pyroptotic tumors and inhibit tumor growth by transforming “cold” tumors into “hot” tumors.

In conclusion, this study demonstrates that ICT effectively suppresses HCC cell growth both in vitro and in vivo. In vitro, ICT inhibited the survival and proliferation of HCC cells and induced typical pyroptotic changes. The caspase1-GSDMD and caspase3-GSDME pathways are involved in ICT-triggered pyroptosis. Additionally, ICT causes HCC cells to undergo pyroptosis in TME, resulting in the release of inflammatory cytokines that transform macrophages into a proinflammatory phenotype. In vivo, ICT treatment notably enhanced the expression of cleaved-caspase1, cleaved-caspase3, and granzyme B, modulated cytokine secretion, and stimulated immune cell infiltration, leading to tumor growth suppression in HCC mice. These findings provide scientific evidence for the application of ICT in the therapy of HCC through the regulation of pyroptosis and immune response.

## Figures and Tables

**Figure 1 biomedicines-12-01917-f001:**
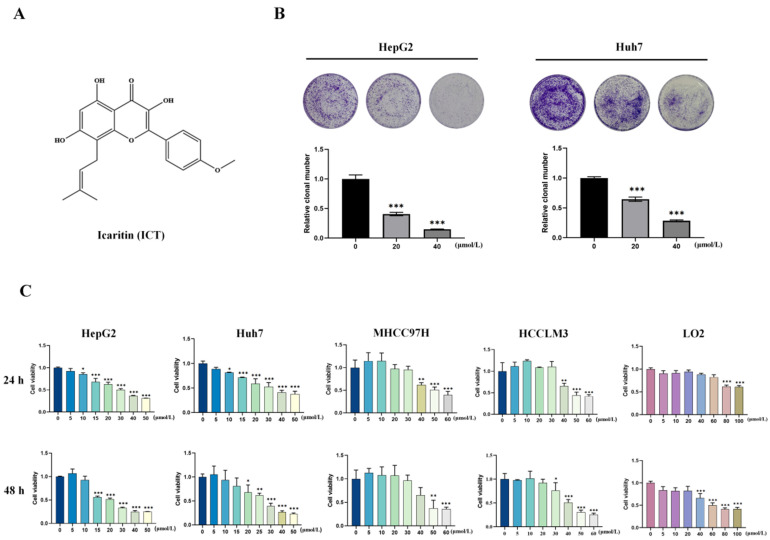
Icaritin (ICT) inhibited cell viability and proliferation of hepatocellular carcinoma (HCC) cells. (**A**) Chemical structural formula of ICT. (**B**) Colony formation in HepG2 and Huh7 cells subjected to ICT (0–40 μM). (**C**) Cell viability of HepG2, MHCC97H, HCCLM3, Huh7, and LO2 cells given ICT for 24 or 48 h. Data are presented as the mean ± SD. The following symbols denote statistical significance: ** p* < 0.05, ** *p* < 0.01, *** *p* < 0.001.

**Figure 2 biomedicines-12-01917-f002:**
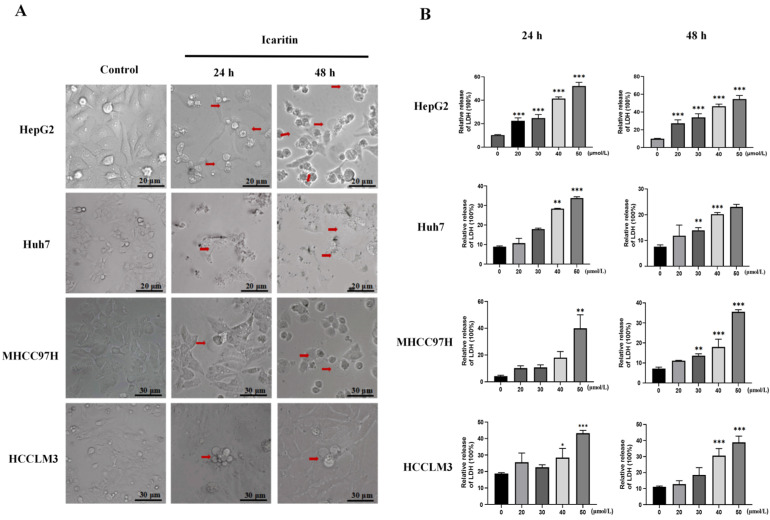
HCC cells underwent pyroptosis caused by ICT. (**A**) Microscopy images showing pyroptosis in HCC cells treated with ICT (40× magnification). Red arrows indicate cells exhibiting large bubbles. (**B**) LDH release in HCC cells subjected to ICT (0–50 μM) for 24 and 48 h. (**C**) Images of HepG2 cells stained with Annexin V-PI after 48 h of ICT (0–25 μM) treatment. Red arrows indicate cells exhibiting large bubbles. (**D**) Flow cytometry pseudo-color dot plots demonstrating annexin V-PI staining in HepG2 cells utilized with ICT (0–25 μM) for 48 h. (**E**) LDH release in HepG2 and Huh7 cells induced by 25 µM ICT and different inhibitors for 48 h. Inhibitors include Z-VAD-FMK (a pan caspase inhibitor), Ac-DEVD-CHO (a caspase-3 inhibitor), and disulfiram (a GSDMD inhibitor). The mean ± SD is displayed in the graphs. * *p* < 0.05, ** *p* < 0.01, *** *p* < 0.001.

**Figure 3 biomedicines-12-01917-f003:**
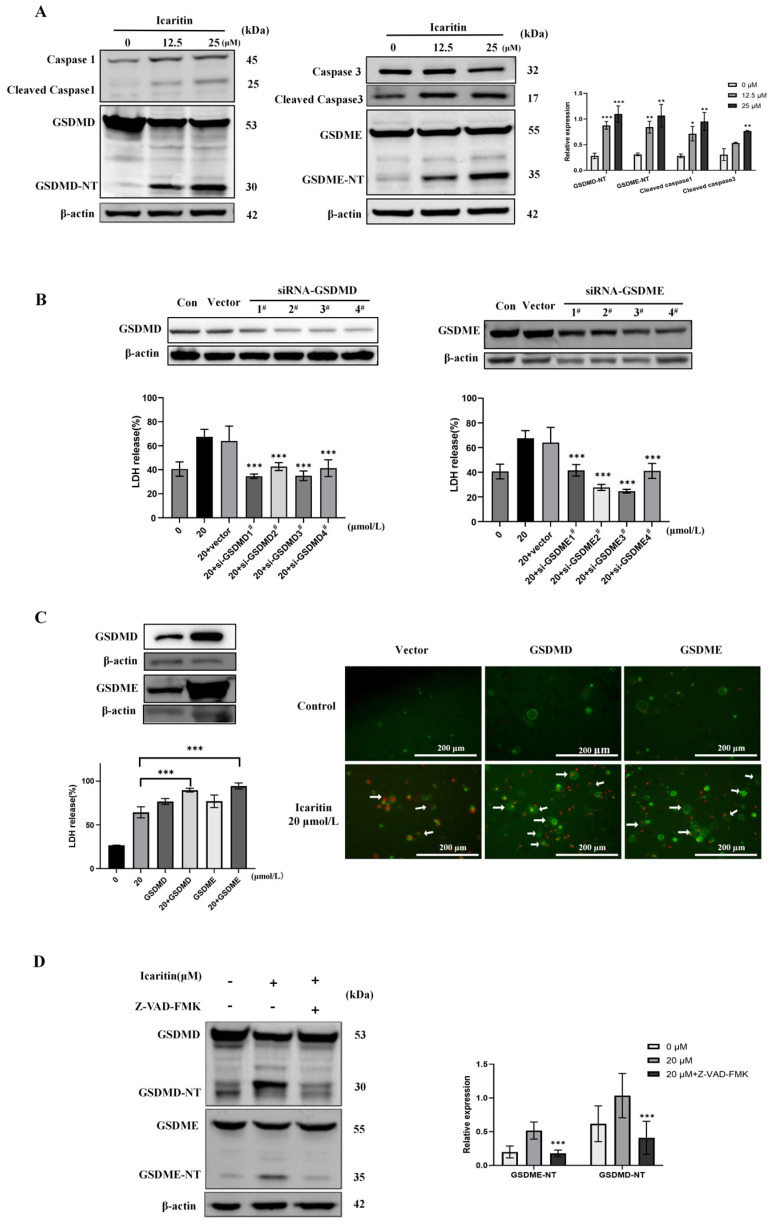
The caspase1-GSDMD and caspase3-GSDME pathways are involved in ICT-triggered pyroptosis. (**A**) Expression levels of cleaved-caspase-1/3 and GSDMD/E-NT in HepG2 cells exposed to 0–25μM ICT for 48 h, assessed using Western blotting. (**B**) LDH release in GSDMD/E knockdown HepG2 cells treated with 20 μM ICT for 48 h. (**C**) LDH release and annexin V-PI staining in GSDMD/E overexpressed HepG2 cells subjected to 20 μM ICT for 48 h. White arrows indicate cells exhibiting large bubbles. (**D**) Expression of GSDMD/E-NT in HepG2 cells stimulated for 48 h with 20 μM ICT and Z-VAD-FMK inhibitor, analyzed using Western blotting. The graphs represent the mean ± SD. * *p* < 0.05, ** *p* < 0.01, *** *p* < 0.001.

**Figure 4 biomedicines-12-01917-f004:**
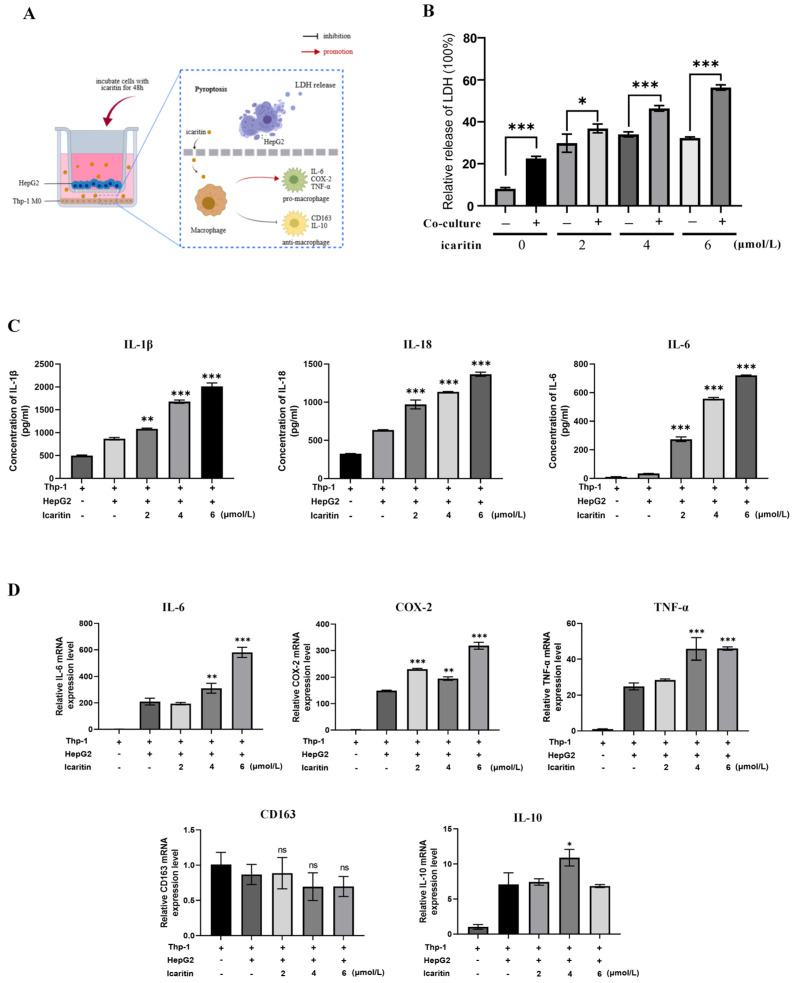
Pyroptosis induced the release of inflammatory factors and promoted macrophage transformation towards a proinflammatory phenotype. (**A**) Experimental diagram of the co-culture system. (**B**) LDH release was detected in the co-culture system treated with 0–6 µM ICT for 48 h. (**C**) ELISA of inflammatory cytokines IL-1β, IL-18, and IL-6. Statistical analyses were conducted to compare the groups of co-cultured HepG2 and THP-1 cells with and without ICT treatment groups. (**D**) Expression levels of proinflammatory macrophage (M1) markers (COX-2, IL-6, TNF-α) and anti-inflammatory (M2) markers (CD163, IL-10) assessed using qPCR. Statistical analyses were conducted to compare the THP-1 cell group with other groups. The graphs represent the mean ± SD. The n.s. represent no significance and *p* < 0.05 was considered statistically significant., * *p* < 0.05, ** *p* < 0.01, *** *p* < 0.001.

**Figure 5 biomedicines-12-01917-f005:**
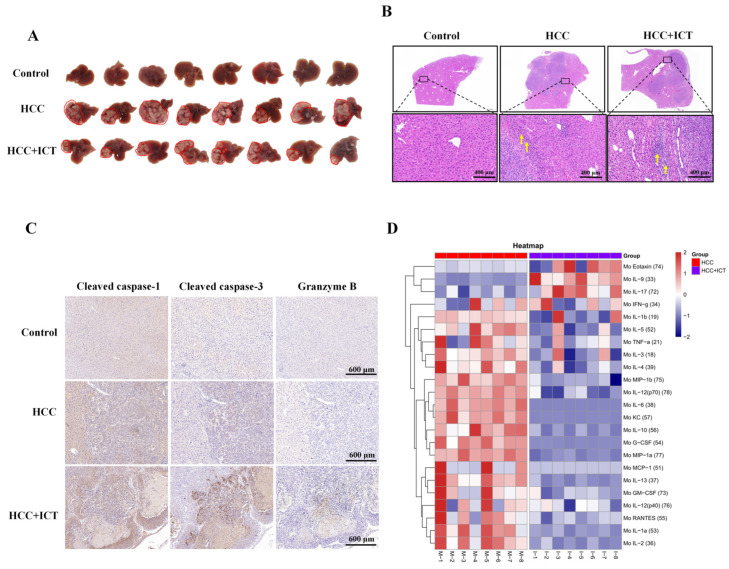
The anti-hepatocarcinoma effects of ICT on the regulation of pyroptosis, cytokines, and immunity. (**A**) Graph demonstrating the significant inhibitory effect of ICT on tumor growth. (**B**) H&E staining of the liver with magnification of cell features; yellow arrows indicate interstitial inflammatory cell infiltration. (**C**) Immunohistochemical staining revealing elevated expression of cleaved-caspase-1/3 and granzyme B in ICT-treated samples, with brown coloration indicating positive staining. (**D**) Differences in cytokine expression between HCC and ICT groups. (**E**) Representative multiplex immunohistochemistry staining depicting the infiltration of various immune cells.

**Table 1 biomedicines-12-01917-t001:** Sequences for quantitative real-time PCR.

Primers	Sequences (5′-3′)
IL-6	F: CGGGAACGAAAGAGAAGCTCTA
IL-6	R: CGCTTGTGGAGAAGGAGTTCA
IL-10	F: TCAAGGCGCATGTGAACTCC
IL-10	R: GATGTCAAACTCACTCATGGCT
CD163	F: CAGCGGCTTGCAGTTTCCTC
CD163	R: TGGCCTCCTTTTCCATTCCAGA
COX-2	F: CCACCCGCAGTACAGAAAGT
COX-2	R: CAGGATACAGCTCCACAGCA
TNF-α	F: GAGGCCAAGCCCTGGTATG
TNF-α	R: CGGGCCGATTGATCTCAGC
β-actin	F: CTCTTCCAGCCTTCCTTCCT
β-actin	R: CAGGGCAGTGATCTCCTTCT

**Table 2 biomedicines-12-01917-t002:** Sequences of siRNA GSDMD and GSDME knockdown.

Primers	Sequences (5′-3′)
SiGSDMD-1	F: GAGCUUCCACUUCUACGAUTT
SiGSDMD-1	R: AUCGUAGAAGUGGAAGCUCTT
SiGSDMD-2	F: GACACAGAAGGAGGUGGAATT
SiGSDMD-2	R: UUCCACCUCCUUCUGUGUCTT
SiGSDMD-3	F: GCCAUCUGAGCCAGAAGAATT
SiGSDMD-3	R: UUCUUCUGGCUCAGAUGGCTT
SiGSDMD-4	F: CCACAACUUCCUGACAGAUTT
SiGSDMD-4	R: AUCUGUCAGGAAGUUGUGGTT
SiGSDME-1	F: GGUGACCUGAUUGCAGUAUTT
SiGSDME-1	R: AUACUGCAAUCAGGUCACCTT
SiGSDME-2	F: GCAGCAAGCAGCUGUUUAUTT
SiGSDME-2	R: AUAAACAGCUGCUUGCUGCTT
SiGSDME-3	F: GGAUUGUGCAGCGCUUGUUTT
SiGSDME-3	R: AACAAGCGCUGCACAAUCCTT
SiGSDME-4	F: GCUGCGCAUGGGAUAUCUUTT
SiGSDME-4	R: AAGAUAUCCCAUGCGCAGCTT

## Data Availability

The datasets used during and/or analyzed during the current study are available from the corresponding author on reasonable request due to privacy and ethical reasons.

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
