# Peer review of "Icaritin Exerts Anti-Cancer Effects through Modulating Pyroptosis and Immune Activities in Hepatocellular Carcinoma"

_biomedicines, 2024, doi:10.3390/biomedicines12081917_

Round 1

Reviewer 1 Report

Comments and Suggestions for Authors

Icaritin is a bioactive compound derived from the Epimedium species, commonly known as horny goat weed. It belongs to a class of flavonoids and is known for its various pharmacological properties, such as anti-cancer, anti-inflammatory, and neuroprotective effects.   In their submitted manuscript, Yuanyuan et al. demonstrate that ICT exhibits pyroptosis-inducing and anti-hepatocar-12 cinoma effects, which is intriguing.  Building on these findings, the authors illustrate ICT promoted pyroptosis in co-cultivation of HepG2 cells and macrophages, regulating the release of inflammatory cytokines and the transformation of macrophages into a proinflammatory phenotype. To me, this work represents a step forward in the understanding of pharmacological mechanism of ICT. I have the following questions/comments.

1.     In Figure 2B, how did the authors perform the LDH release assay? Why is the LDH release in HepG2 at 24 hours over 100%? Does this result make sense? Please explain it in detail.

2.     In Figure 3B, what are the effects of simultaneously knocking down GSDMD and GSDME using siRNA?

3.     Between GSDMD and GSDME, which is predominantly expressed in HepG2 cells? Please test this using qPCR.

4.     The method section requires substantial improvement to enhance its clarity and reproducibility for other researchers.

5.     Some modifications/corrections in the manuscript may be needed.

Comments on the Quality of English Language

N/A

Author Response

Dear reviewer,

         Thank you very much. We appreciate your insightful comments and suggestions. We have carefully considered all your suggestions and have revised the manuscript accordingly. Below is a point-by-point response to each of your comments, please see the attachment. Thank you once again for your consideration. Wish you all the best in your work and life.

Reviewer 2 Report

Comments and Suggestions for Authors Yuanyuan Jiao and colleagues investigate Icaritin anti-cancer effect in hepatocellular carcinoma. They clearly show that Icaritin can induce pyroptosis on these cancer cells. However, their results concerning the immune response induce by Icaritin treatment should be improved.      Line 244 the authors state about their results "This confirmed the novel role of ICT in a switch from apoptosis to pyroptosis (Figure 2D) »  Since LDH release, cellular morphology and Annexin V stainning can be associated to apoptosis we can also point out that pyroptosis and apoptosis occur after ICT treatment. Analyses of GSDMD and GSDME clearly indicate the induction of pyroptosis after ICT treatment.   How the authors could explain the increase of LDH release after their coculture conditions ? figure 4B Are these effects also observed when icaritin is directly added on a culture of macrophages (Thp1 M0) alone? This point should be adressed. The authors should clarify if ICT modulate cytokines release from cancer cells or macrophages ?    For the in vivo part, a graphic representing the tumor volume should be added. How the authors could explain the decrease of pro-inflamatory cytokines whereas they observe the inverse in vitro when macrophages are cocultured with cancer cells?    

Author Response

(The authors gave the same response as above.)

Reviewer 3 Report

Comments and Suggestions for Authors

The work presented by Jiao and colleagues is of great interest.

The use of natural products is often elusive when it comes to identifying molecular pathways.

The work requires some English language revision:

LANE 41;

can emit granzymes (Gzms) that activate gasdermins (GSDMs) to 41; the word EMIT can be replaced by secretion for example.

LANE 342

nating from the plasma membrane, were observed in ICT-induced HepG2, Huh7, 342; can be replaced by ICT-treated or rephrased.

The introduction can be slightly more comprehensive.  Pyroptosis is a new conceptual cell death program, discovered 20 years ago. GSDMs were first reported in 2007. In particular, what is the difference between GSDM A, B, C, D, E and hPJVK. That will help the reader to identify what is expected. Not all GSDMs are expressed in all tissues.

In the Discussion, caspases are identified alongside Granzyme B, but GSDMs are not. Interesting will be if the correlation found in vitro is the same. The Physiology of mice seems different since for example they lack GSDM-B(https://www.ncbi.nlm.nih.gov/pmc/articles/PMC10164276/).

Were all GSDMNs examined in vitro or only D and E? Were granzymes examined in vitro? What is the difference between some immortalized cells being more susceptible to ICT treatment?

Figure 5D, is too small to read. Perhaps a better description in the legend and an expanded mention in the "Results" could help the reader better understand the data.

Figure 5E, the units are "mm" is that correct or it should be "nanometer (nm)"

References are needed :

In HCC cells, NK cells and CD8+ Cytotoxic 40

T Lymphocytes (CTLs) can emit granzymes (Gzms) that activate gasdermins (GSDMs) to 41

induce pyroptosis. (HERE)

Icaritin (ICT), a highly potent 55 (HERE also) Potency is not well defined for ICT or in comparison to other natural products or therapeutics.

DOES ICT enter the cell with assistance from proteins (receptors or co-receptors)? Is water soluble?

What is the significance of Six-week-old male mice? Is age relevant? Is sex relevant?

Finally, in vivo, HEPG cells were selected. Were there other cell lines tested or that could have given similar results? In figure 1 several other cells were tested: f HepG2, MHCC97H, HCCLM3, Huh7 and LO2.

Comments on the Quality of English Language

It was mentioned in the main review some minor details.

Author Response

(The authors gave the same response as above.)

Round 2

Reviewer 1 Report

Comments and Suggestions for Authors

The authors have answered my questions. I don't have other questions.